# The Effects of Nicotine Pouches and E-Cigarettes on Oral Microbes: A Pilot Study

**DOI:** 10.3390/microorganisms12081514

**Published:** 2024-07-24

**Authors:** Sintija Miluna-Meldere, Dagnija Rostoka, Renars Broks, Kristine Viksne, Rolands Ciematnieks, Ingus Skadins, Juta Kroica

**Affiliations:** 1Department of Prosthetic Dentistry, Riga Stradins University, LV-1007 Riga, Latvia; sintijamiluna@gmail.com; 2Department of Biology and Microbiology, Riga Stradins University, LV-1007 Riga, Latvia; renars.broks@rsu.lv (R.B.); ingus.skadins@rsu.lv (I.S.); juta.kroica@rsu.lv (J.K.); 3Institute on Oncology and Molecular Genetics, Riga Stradins University, LV-1007 Riga, Latvia; kristine.viksne@rsu.lv (K.V.); rolands.ciematnieks@rsu.lv (R.C.)

**Keywords:** pathogenic microorganisms, nicotine pouches, electronic cigarettes, smokeless tobacco, saliva

## Abstract

It remains uncertain whether nicotine pouches and electronic cigarettes alter the oral environment and result in a high presence of periodontopathogenic bacteria in saliva, compared to that among cigarette users or non-tobacco users. In this study, saliva samples were collected from respondents using nicotine pouches, electronic cigarettes, and conventional cigarettes, alongside a control group of non-tobacco users. Polymerase chain reaction was used to identify clinical isolates of the following periodontal bacteria: *Porphyromonas gingivalis*, *Tannerella forsythia*, *Prevotella intermedia*, *Fusobacterium nucleatum*, *Fusobacterium periodonticum*, *Porphyromonas endodontalis*, and *Rothia mucilaginosa.* The presence of some periodontal pathogens was detected in the saliva samples from users of nicotine pouches, electronic cigarettes, and conventional cigarettes but not in samples taken from the control group. Therefore, the initial results of this pilot study suggest that the presence of periodontopathogenic bacteria in the saliva of nicotine pouch and electronic cigarette users could alter the oral microbiome, leading to periodontal diseases. However, further quantitative investigation is needed.

## 1. Introduction

The oral microbiota plays a significant role in the human body and affects other parts of the body, including the gut microbiota [1]. Recent research has found that dysbiosis of the oral microbiota can lead to the development of oral cancer [2,3,4], potentially due to higher levels of periodontal pathogens and fungi [5].

Tobacco use is also a significant risk factor for the development of cancer [6]. Novel tobacco products, such as electronic cigarettes and nicotine pouches, are currently gaining popularity among children and young adults, surpassing traditional cigarettes [7]. However, the potential consequences of these products on oral health remain unclear. There is strong evidence that cigarette users have a higher prevalence of periodontal pathogens [8,9], but literature studying the impact of nicotine pouches and electronic cigarettes on the oral microbiota are limited [10,11]. Some studies have explored the impacts of smokeless tobacco on the oral microbiota [12,13], but these studies were conducted on Sudanese or Indian smokeless tobacco, which is not used in Europe. Instead, Swedish snus is typically selected by European consumers [14,15].

Nicotine pouches are a non-combustible nicotine product that is contained in small pouches and used by placing the pouch under the upper or lower lip in the oral cavity. Based on its appearance and usage, this product looks identical to Swedish snus [16]. However, Swedish snus contains tobacco and is legally sold only in Sweden, not European Union countries [17]. Nicotine pouches legally sold in markets throughout the European Union do not contain tobacco and are advertised as a new alternative to combustible tobacco products. This type of product attracts people who have never used tobacco or nicotine products before, increases addiction, and influences poor oral and systemic health [18]. Pouches offer different flavors, with nicotine content ranging from 6 mg/g per pouch up to 50 mg/g per pouch. Nicotine pouches also contain many potentially harmful substances [19]. Manufacturers advise using this product for 30 min [20], but the average duration of use is 75 min [21]. This product is gaining in popularity because it is so easy to use and lacks an odor, which could affect others in public places. This product’s possible harmful effects on oral health, however, are still unclear. These pouches began to be sold on the European market in 2019, during the time when coronavirus disease 2019 (COVID-19) was active and people were not allowed to go outside [22].

Electronic cigarettes have been available for many years and are currently the most popular nicotine-based product among young people and adolescents, possibly because of the disposable electronic options that come in many flavors and colors, making them both trendy and affordable [23]. Moreover, the number of people interested in these products has been increasing every year [24]. This growing interest is particularly notable among individuals who have never previously shown interest in tobacco or nicotine products [25]. Additionally, approximately 29% of adults now consume dual-use products, such as electronic and conventional cigarettes, daily [26]. These consumption habits also raise questions about how multiple tobacco/nicotine products influence oral and systemic health.

Current data indicate that electronic cigarette users exhibit greater alpha diversity in their oral microbiota [27], suggesting oral dysbiosis and the potential risk of periodontal disease [28]. Moreover, red-complex bacteria, such as *Porphyromonas gingivalis* and *Fusobacterium nucleatum*, have been linked to the development of oral squamous cell carcinoma [29,30,31]. Indeed, the genera *Lactobacillus* and *Treponema* have been found in the oral cavities of patients with smoking-related oral squamous cell carcinoma [32].

Research by Cicchinelli et al. showed that healthy smokers have elevated levels of *Fusobacterium nucleatum*, *Prevotella intermedia*, and *Prevotella nigrescens* in their oral cavities. In contrast, ill smokers (those with periodontitis) have elevated levels of *Porphyromonas gingivalis*, *Tannerella forsythia*, and *Treponema denticola*, which not only cause dysbiosis in the oral cavity but also affect other organs of the body [8]. Tobacco and nicotine products disrupt the oral microbiota, favoring pathogenic microorganisms that lead to inflammation and disease [33]. Electronic cigarette vapor is believed to increase the diversity of the microbiota, with the saliva microbiome being affected less than the subgingival microbiome [28]. Moreover, vapor suppresses the growth of commensals while enhancing opportunistic pathogens [34].

Under this background, it is important to understand how nicotine pouches and Swedish snus impact periodontal pathogens, as both may play a significant role in future disease development. Moreover, saliva is an easily accessible and patient-friendly method for diagnosing pathogenic microorganisms that can be utilized as a screening tool for potential pathogen changes before more serious and financially expensive detection methods are employed.

Therefore, the aim of the present study was to compare the presence of periodontal pathogens in the saliva of nicotine pouch users, electronic cigarette users, and conventional cigarette users to their presence in the saliva of respondents who do not use tobacco and/or nicotine products. We hypothesized that periodontal pathogens would be present in the saliva of nicotine pouch and electronic cigarette users but not in the control group (non-tobacco/non-nicotine users).

## 2. Materials and Methods

### 2.1. Study Design

This pilot study is part of a larger research project approved by the Ethics Committee of Riga Stradins University (No. 22/28.01.2016). Participation in this study was voluntary, and participants signed informed consent forms on the collection of biological material and participation in the study. All procedures adhered to the Declaration of Helsinki.

Participants (n = 60) completed online questionnaires on their tobacco/nicotine product consumption (duration of tobacco/nicotine product use, daily units used, storage conditions, age at initiation, addiction onset age, willingness to quit, etc.), medical health (systemic diseases or medical conditions, alcohol units used per day, daily medication intake, allergies, last antibiotic use, etc.), oral health (teeth cleaning habits, dental floss usage, last dental visit, last visit to a dental hygienist, presence of removable/fixed dentures, etc.), and dietary habits (food-related allergies, no current long-term diets such as keto or fresh juice diet, etc.). Sociodemographic questions were also asked (age, sex, ethnicity, education, current work status, etc.). 

The inclusion criteria were as follows: age 18–35; no systemic diseases or medical conditions; no daily alcohol intake; the absence of a current pregnancy; the absence of caries; the absence of periodontitis; the absence of daily medication intake; the absence of removable/fixed dentures; the absence of antibiotic usage for at least a year; and the use of one tobacco or nicotine product daily for at least 2 years. Based on these criteria, 45 respondents were selected to continue the study, with an oral examination performed by a certified dentist. 

The oral examination consisted of assessing dental status, conducting a basic periodontal examination (BPE), and examining the oral mucosa. Participants with a basic periodontal examination score of 3 or 4, active caries, or removable/fixed dentures were excluded from the study (n = 14). In total, 31 respondents continued with the study. 

### 2.2. Collection of Saliva Samples

Participants were instructed not to eat, brush their teeth, drink, or use tobacco/nicotine products for 30 min before the oral examination and saliva sample collection. Saliva samples (5 mL) were collected and placed in Eppendorf tubes (5 mL, Sigma-Aldrich, Hamburg, Germany).

### 2.3. Genomic DNA Preparation

Thawed saliva samples were transferred to Lysing Matrix E tubes (MP Biomedicals, Irvine, CA, USA) and homogenized using a FastPrep-24™ bead-beating system (MP Biomedicals, Irvine, CA, USA) at a speed setting of 6.0 for 40 s. The samples were then centrifuged at 14,000× *g* for 10 min, and the supernatant was used for DNA extraction via the phenol–chloroform method [35]. DNA quantity was assessed using a Qubit dsDNA HS Assay Kit and a Qubit 2.0 Fluorometer (Thermo Fisher Scientific, Waltham, MA, USA).

### 2.4. Polymerase Chain Reaction (PCR)

Eight previously referenced PCR primers were used to identify clinical isolates of periodontal bacteria (Table 1). PCR was performed following the corresponding reference methods to detect periodontal bacteria in the isolated DNA. PCR was conducted using a FastGene Ultra Cycler Gradient FG-TC01 (Nippon Genetics, Tokyo, Japan), and the results were visualized in 1.5% agarose gel electrophoresis stained with the Midori Green Advance DNA stain.

### 2.5. Statistics

Due to the small sample size and uneven distribution of the respondents, conducting a meaningful statistical analysis was not feasible. Therefore, only descriptive statistics were used. 

## 3. Results

In total, 31 saliva samples were collected: 16.1% were control samples (n = 5), 12.9% were from cigarette users (n = 4), 29.0% were from electronic cigarette users (n = 9), and 41.9% were from nicotine pouch users (n = 13). 

The sociodemographic and clinical characteristics of the respondents are presented in Table 2. The sociodemographic characteristics of the respondents revealed a diverse age range, spanning from 19 to 29 years old, with representation from both sexes. Among the total respondents, 23 were men, and 8 were women, all of whom identified as northern Europeans. In terms of educational attainment, 15 respondents completed secondary education, while 16 achieved higher education qualifications.

The duration of tobacco/nicotine use varied significantly across different groups within the study. In the cigarette group, all respondents reported using cigarettes for a period of 5 to 10 years (n = 4). In the electronic cigarette group, all participants indicated using electronic cigarettes for the same duration of 5 to 10 years (n = 9). In contrast, among those using nicotine pouches, 2 respondents reported using the product for less than 5 years, while 11 respondents reported using nicotine pouches for a duration of 5 to 10 years.

The daily consumption of tobacco/nicotine units varied significantly among the different groups studied. In the cigarette group, three respondents reported smoking 10–20 cigarettes per day, with one participant smoking more than 20 cigarettes daily. Within the electronic cigarette group, the majority (n = 8) typically used 1–2 pods per week, while only one respondent reported using more than 2 pods weekly. Conversely, in the nicotine pouch group, half of the respondents consumed 5–10 pouches daily, whereas six respondents reported using more than 10 pouches per day. 

The clinical examination findings indicated varying levels of periodontal health among the different groups studied. In the control group, BPE scores predominantly showed minimal periodontal involvement, with three participants scoring “1” and two participants scoring “2”. Among the cigarette group, we observed mixed periodontal health statuses: one participant scored “1” on the BPE, while eight participants scored “2”, indicating a higher degree of periodontal involvement, compared to that in the control group. In contrast, the nicotine pouch group exhibited a different distribution of BPE scores: 2 participants scored “1”, suggesting minimal periodontal involvement, while 11 participants scored “2”, indicating more significant periodontal issues, compared to those in both the control and cigarette groups.

Interestingly, oral mucosal changes were exclusively observed within the nicotine pouch group, affecting five respondents. These changes were localized at the areas where nicotine pouches were typically placed. The mucosal changes varied in size among the affected individuals within the nicotine pouch group, yet they consistently presented as white and leathery in appearance.

The positive presence of *Porphyromonas gingivalis* was detected in the saliva of both electronic cigarette users (n = 1) and nicotine pouch users (n = 2). *Porphyromonas gingivalis* was not present in the saliva samples of control respondents (n = 5), cigarette users (n = 4), electronic cigarette users (n = 8), or nicotine pouch users (n = 11). 

The positive presence of *Tannerella forsythia* was detected in the saliva of cigarette users (n = 4), electronic cigarette users (n = 5), and nicotine pouch users (n = 11). *Tannerella forsythia* was not present in saliva samples from the control respondents (n = 5), electronic cigarette users (n = 4), or nicotine pouch users (n = 2). All samples from cigarette users tested positive for the presence of *Tannerella forsythia*. 

The positive presence of *Prevotella intermedia* was detected in the saliva of electronic cigarette users (n = 1) and nicotine pouch users (n = 2). *Prevotella intermedia* were not present in the saliva samples from the control respondents (n = 5), cigarette users (n = 4), electronic cigarette users (n = 8), or nicotine pouch users (n = 11). 

The positive presence of *Fusobacterium nucleatum* was detected in the saliva of the control sample (n = 1), cigarette users (n = 4), electronic cigarette users (n = 8), and nicotine pouch users (n = 8). *Fusobacterium nucleatum* was not present in the saliva samples from the control respondents (n = 4), electronic cigarette users (n = 1), or nicotine pouch users (n = 5). All samples from cigarette users tested positive for the presence of *Fusobacterium nucleatum*. 

The positive presence of *Porphyromonas endodontalis* was detected in saliva of cigarette users (n = 3), electronic cigarette users (n = 4), and nicotine pouch users (n = 8). *Porphyromonas endodontalis* was not present in the saliva samples of control respondents (n = 5), cigarette users (n = 1), electronic cigarette users (n = 5), or nicotine pouch users (n = 5). 

The positive presence of *Rothia mucilaginosa* was detected in the saliva of the control sample (n = 3), cigarette users (n = 4), electronic cigarette users (n = 9), and nicotine pouch users (n = 13). *Rothia mucilaginosa* was not present in the control group samples (n = 2). All samples from cigarette, electronic cigarette, and nicotine pouch users tested positive for the presence of *Rothia mucilaginosa*. 

The positive presence of *Fusobacterium periodonticum* was detected in the saliva of the control sample (n = 3), cigarette users (n = 4), electronic cigarette users (n = 6), and nicotine pouch users (n = 12). *Fusobacterium periodonticum* was not present in the saliva samples from control respondents (n = 2), electronic cigarette users (n = 3), or nicotine pouch users (n = 1). All samples from cigarette users tested positive for the presence of *Fusobacterium periodonticum*. 

## 4. Discussion

The results of our pilot study indicate that certain periodontal pathogens (*Tannerella forsythia*, *Prevotella intermedia*, and *Porphyromonas gingivalis*) were present in saliva samples taken from users of nicotine pouches, electronic cigarettes, and conventional cigarettes but not in those taken from the control group. Therefore, the hypotheses of our study were partially confirmed. Despite the small and uneven distribution of respondents across groups, there was a notable positive trend in the results, suggesting an association between the use of tobacco and nicotine products and the presence of certain periodontal pathogens. For example, *Tannerella forsythia* was detected in saliva from the majority of respondents in the nicotine pouch group (n = 11), the majority of respondents in the electronic cigarette group (n = 5), and all respondents in the cigarette group (n = 4).

In some cases, *Prevotella intermedia* and *Porphyromonas gingivalis* were found in the saliva of electronic cigarette users and nicotine pouch users but not in the control group or cigarette group. After a thorough review of the respondents’ results, we found that these outcomes could be attributed to the prolonged and high-dosage use of nicotine products by these particular respondents. *Prevotella intermedia* and *Porphyromonas gingivalis* were present in the saliva of electronic cigarette users who consumed more than two pods per week for nine years and nicotine pouch users who used more than ten pouches per day for more than seven years in a row. 

All respondents from the cigarette group showed the positive presences of *Porphyromonas gingivalis*, *Tannerella forsythia*, *Prevotella intermedia*, *Fusobacterium nucleatum*, *Rothia mucilaginosa*, and *Fusobacterium periodonticum* in their saliva samples. The results of our study are consistent with previous findings indicating that cigarette users more commonly harbor periodontopathogenic bacteria in their saliva than non-tobacco users [42]. *Tannerella forsythia* and *Fusobacterium nucleatum* are well-known periodontal pathogens commonly found in patients with periodontitis [43]. *Fusobacterium nucleatum* exacerbates inflammatory responses, promotes tumor progression, damages periodontal tissues, and facilitates cell invasion [44]. *Tannerella forsythia* is recognized for its proteolytic activity, which contributes to periodontal tissue damage [45]. *Porphyromonas gingivalis* elevates IL-8, TNF-alpha, IL-1, and IL-6 expressions in gingival epithelial cells, inhibits stem cell proliferation and differentiation [46,47], and negatively affects periodontal ligament fibroblasts, particularly in the presence of nicotine [48]. Based on these findings, cigarette use strongly influences the presence of periodontopathogenic bacteria in one’s saliva.

In contrast, the results from the electronic cigarette users were less conclusive. While certain periodontopathogenic bacteria were detected in the majority of cases among users of electronic cigarettes (for example, *Fusobacterium nucleatum* and *Fusobacterium periodonticum*), *Porphyromonas gingivalis* was predominantly absent. This discrepancy may be attributable to variations in bacterial affinity for nicotine among different species, as well as the selective modulation of bacterial growth by nicotine [42]. As Thomas et al. concluded, electronic cigarette users promote a unique oral microbiome that exhibits some characteristics typical of conventional cigarette users and some characteristics typical of non-tobacco users [49], thereby explaining why certain typical periodontopathogens were absent in the saliva samples. Additionally, tobacco consumption alters the oral environment, favoring the growth of some bacteria while inhibiting others [50]. Xu et al. concluded that the oral microbiota of electronic cigarette users exhibits similarities to that of cigarette users, as the use of electronic cigarettes increases pro-inflammatory cytokine levels in saliva, leading to oral dysbiosis [51]. Nevertheless, since electronic cigarettes are a relatively new product on the market, further studies are needed to gather comprehensive information about changes in the oral microbiome due to exposure to electronic cigarette vapor.

The majority of nicotine pouch users showed the positive presence of *Tannerella forsythia* in their saliva samples, while *Porphyromonas gingivalis* and *Prevotella intermedia* were mostly absent. Interestingly, the presences of *Fusobacterium nucleatum* and *Porphyromonas endodontalis* were comparable in the saliva samples. Similarities were also observed with waterpipe users [52]. Moreover, systemically healthy individuals with inflammation tended to have higher proportions of periodontopathogenic bacteria in the oral cavity, compared to those with severe periodontitis [53]. Since our study excluded individuals with present periodontitis or a basic periodontal examination score of 3 or 4, the positive presences of some periodontopathogenic bacteria in the saliva may suggest changes in the oral environment that eventually lead to gum disease.

In the control group, most periodontal pathogens were absent from the saliva samples (*Porphyromonas gingivalis*, *Tannerella forsythia*, *Prevotella intermedia*, and *Porphyromonas endodontalis*), except for *Fusobacterium nucleatum*, which was detected in one respondent. Typically, *Fusobacterium nucleatum* is found in the saliva of individuals with gingivitis or periodontitis, rather than in healthy individuals [54,55]. Some respondents in the control group showed the positive presences of *Rothia mucilaginosa* and *Fusobacterium periodonticum*, but all respondents in the tobacco or nicotine groups had positive presences of these bacteria. *Rothia mucilaginosa* is a commensal bacterium found in healthy oral cavities without caries [56], explaining the positive presence of this bacterium in the control group. *Rothia mucilaginosa* has also been found in abundance in tongue leukoplakia lesions [3]. Furthermore, *Rothia mucilaginosa* not only acts as an effective nitrate reducer [57] but also serves as a biomarker for halitosis-free patients [58]. Additionally, since *Rothia mucilaginosa* is an acetaldehyde-producing bacterium, its presence may contribute to the initiation of oral cancer, especially when combined with tobacco and alcohol consumption [59,60]. *Fusobacterium periodonticum* was present in all saliva samples from cigarette users but was mostly found in samples from electronic cigarette and nicotine pouch users. Increased levels of *Fusobacterium periodonticum* have been associated with oral squamous cell carcinoma [61], while decreased levels have been linked to head and neck squamous cell carcinoma [62]. 

Some intriguing patterns were identified among nicotine pouch users based on the questionnaire responses. It was noted that respondents using nicotine pouches often consume more than one pouch simultaneously. This practice significantly increases nicotine intake, especially for those using 5–10 pouches per day or more than one pouch at a time. Consequently, daily nicotine uptake among these users can be substantial, highlighting potential concerns regarding nicotine consumption levels [63]. Moreover, some respondents revealed that nicotine pouches were sometimes used during sports training sessions. It is known that nicotine can potentially enhance stimulation, increase alertness, and improve coordination among users. The World Anti-Doping Agency (WADA) included nicotine in its 2013 monitoring program but has not banned the drug’s use among athletes [64]. Conversely, some respondents used pouches during sleep, risking choking and the possible development of dental caries, as saliva flow is reduced during the night.

Furthermore, nicotine pouch users exhibited oral mucosal changes characterized by white, leathery patches observed at the sites where pouches were regularly placed. Traditionally, such oral mucosal alterations have been documented predominantly among users of smokeless tobacco products [65,66]. However, research specifically focusing on nicotine pouches remains limited in literature. Studies have indicated that the use of smokeless tobacco could lead to leukoplakia [67], oral cell dysplasia [68], parakeratosis, and hyperkeratosis [69]. Although leukoplakia can be clinically diagnosed, biopsies are necessary to obtain definitive results. Biopsies of such oral lesions reveal parakeratosis with acanthosis, which is a histopathological feature of leukoplakia [70]. It is known that oral epithelial dysplasia and other changes can lead to oral squamous cell cancer [71]. Therefore, oral lesions should be monitored by dentists or other medical professionals. 

Based on the questionnaire results for electronic cigarette users, almost all respondents started using electronic cigarettes without any previous tobacco or nicotine history. This result indicates that such products are highly appealing to those who were previously non-tobacco and non-nicotine users. Some respondents admitted that they began using these products because their friends and family used them at gatherings. Unlike traditional cigarettes, electronic cigarettes do not produce unpleasant odors. Instead, their vapor is often described as having a pleasant aroma and taste. Therefore, medical practitioners should inform young people about the harmful effects on health and the potential for addiction arising from the use of electronic cigarettes.

Additionally, there are some challenges in accurately measuring the usage of electronic cigarettes. Unlike conventional cigarettes, which are easy to quantify, there are different classification possibilities for electronic cigarettes, such as counting the number of pods used per week, number of puffs per day, or the amount of nicotine liquid consumed [72]. Therefore, future research should also focus on identifying accurate methods for measuring the use of electronic cigarettes. The limitations of our study include the relatively small number of respondents per group, the uneven distribution of respondents, and the lack of further division based on the frequency of product use.

Overall, the initial results of this pilot study suggest that the presence of periodontopathogenic bacteria in the saliva of nicotine pouch and e-cigarette users could alter the oral microbiome, leading to periodontal diseases. However, further quantitative investigations are needed to confirm these outcomes.

## Figures and Tables

**Table 1 microorganisms-12-01514-t001:** Specific oligonucleotides used in this study.

Bacteria		Sequences (5′-3′)	References
*Porphyromonas gingivalis*	Forward	AGG CAG CTT GCC ATA CTG CG	[36]
	Reverse	ACT GTT AGC AAC TAC CGA TGT	
*Tannerella* *forsythia*	Forward	GCG TAT GTA ACC TGC CCG CA	[36]
	Reverse	TGC TTC AGT GTC AGT TAT ACC T	
*Prevotella intermedia*	Forward	CGA ACC GTC AAG CAT AGG C	[37]
	Reverse	AAC AGC CGC TTT TAG AAC ACA A	
*Fusobacterium nucleatum*	Forward	CGC AGA AGG TGA AAG TCC TGT AT	[38]
	Reverse	TGG TCC TCA CTG ATT CAC ACA GA	
*Fusobacterium periodonticum*	Forward	ACC TTA TCA AGA CTT ATT ATT TC	[39]
	Reverse	TCA AAC TCT ATY TCA GGA ACA A	
*Porphyromonas endodontalis*	Forward	CTA TAT TCT TCT TTC TCC GCA TGG AGG AGG	[40]
	Reverse	GCA TAC CTT CGG TCT CCT CTA GCA TAT	
*Rothia mucilaginosa*	Forward	GCC TAG CTT GCT AGG TGG AT	[41]
	Reverse	GCA GGT ACC GTC AAT CTC TC	

**Table 2 microorganisms-12-01514-t002:** Findings on the sociodemographic and clinical characteristics of the respondents in the study.

Sociodemographic Characteristics	Control Group (n = 5)	Cigarette Group (n = 4)	E-Cigarette Group (n = 9)	Nicotine Pouch Group (n = 13)
Age	21–26	23–29	19–23	26–29
Sex	women, n = 2 men, n = 3	women, n = 0 men, n = 4	women, n = 6 men, n = 3	women, n = 0 men, n = 13
Ethnicity	all northern European	all northern European	all northern European	all northern European
Education	secondary education, n = 2 higher education, n = 3	higher education, n = 4	secondary education, n = 8 higher education, n = 1	secondary education, n = 5 higher education, n = 8
The duration of tobacco/nicotine use	N/A	<5 years, n = 0 5–10 years, n = 4 >10 years, n = 0	<5 years, n = 0 5–10 years, n = 9 >10 years, n = 0	<5 years, n = 2 5–10 years, n = 11 >10 years, n = 0
The daily dose of tobacco/nicotine units	N/A	<10 cigarettes, n = 0 10–20 cigarettes, n = 3 >20 cigarettes, n = 1	<pod per week, n = 0 1–2 pods per week, n = 8 >2 pods per week, n = 1	<5 pouches, n = 0 5–10 pouches, n = 7 >10 pouches, n = 6
**Clinical characteristics**				
BPE	score 1, n = 3 score 2, n = 2	score 2, n = 4	score 1, n = 1 score 2, n = 8	score 1, n = 2 score 2, n = 11
Oral mucosal changes	not observed	not observed	not observed	observed, n = 5 not observed, n = 8

## Data Availability

The original contributions presented in the study are included in the article, further inquiries can be directed to the corresponding author.

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
