# Peer review of "The Effects of Nicotine Pouches and E-Cigarettes on Oral Microbes: A Pilot Study"

_microorganisms, 2024, doi:10.3390/microorganisms12081514_

Round 1

Reviewer 1 Report

Comments and Suggestions for Authors

The purpose of the present study was to compare the presence of periodontal pathogens in the saliva of nicotine pouch users, electronic cigarette users, and conventional cigarette users with their presence in the saliva of respondents who do not use tobacco and/or nicotine products. 

Just one dentist performed the evaluation?

The sudy does not have a conclusion.

Just a descriptive statistics from my point of view is not enough.Could the authors perform also some tests?

The article is well written.

The studture spects all the norms.

I wouls suggest the authors in the discussion section to include more recent published articles.

Comments on the Quality of English Language

Moderate

Author Response

Response to Reviewer 1 comments for research article ,,The Effects of Nicotine Pouches and E-Cigarettes on Oral Microbes: A Pilot Study”

Thank you very much for taking the time to review this manuscript. Please find the detailed responses below and the corresponding corrections highlighted in the re-submitted files.

Here are our responses/ corrections for your comments:

  1. Comment 1: Just one dentist performed the evaluation?
  2. Response1: Yes, oral examination was performed by one certified dentist.

  1. Comment 2: The study does not have a conclusion.
  2. Response 2: Thank you for your comment. The conclusions are located in the last paragraph of the Discussion section (Lines 221-224). According to the journal guidelines, a separate Conclusions section is not mandatory and may be added only if the Discussion is particularly lengthy.

  1. Comment 3: Just a descriptive statistics from my point of view is not enough. Could the authors perform also some tests?
  2. Response 3: Due to the small sample size and uneven distribution of respondents, conducting meaningful statistical analysis was not feasible. Therefore, only descriptive statistics were used. After consulting with a statistics laboratory, we were advised that analysis with such a small sample size would not be meaningful.

  1. Comment 4: I would suggest the authors in the discussion section to include more recent published articles.
  2. Response 4: Thank you for your comment. We have taken it into account and replaced some older articles with newer ones. Please review the track changes in the resubmitted document. Currently, there are no articles older than 9 years (3 articles), and there are 14 articles from the last 5 years in the Discussion section.

Reviewer 2 Report

Comments and Suggestions for Authors

Reviewer suggestions and comments

The authors in this study monitored whether nicotine pouches and electronic cigarettes could alter the oral environment which resulted in high presence of periodontopathogenic bacteria in saliva compared with non-tobacco users. Saliva samples were collected from respondents using nicotine pouches, electronic cigarettes, conventional cigarettes, and a control group of non-tobacco users. Polymerase chain reaction was used to identify clinical isolates of periodontal bacteria, the study suggested that some periodontal pathogens was detected in the saliva samples of users of nicotine pouches, electronic cigarettes, and conventional cigarettes, but not in the control group. Thus, the preliminary findings of this pilot study imply that periodontopathogenic bacteria may change the oral microbiome and cause periodontal diseases if found in the saliva of nicotine pouch and electronic cigarette users.

Overall, the manuscript was good, however, a few major concerns/comments needed to be explained or modified.

  1. Line 125-126 These numbers are few so how can the authors explain it? Did the authors ask them for how long they are addicted to it
  2. Line 152 what was the meaning of positive presence
  3. Comments for table 2 The authors can add alcohol intake and also biochemical test is needed to correlate the study, more data was needed.
  4. First paragraph of discussion “It would be nice if the authors could discuss with numbers and the presence of harmful microorganisms”
  5. Line 184-187 Explain with another published article, try to cite more and comprehensively discuss.
  6. Line 207-208 what would be the possible reason for this

Author Response

Response to Reviewer 2 comments for research article ,,The Effects of Nicotine Pouches and E-Cigarettes on Oral Microbes: A Pilot Study”

Thank you very much for taking the time to review this manuscript. Please find the detailed responses below and the corresponding corrections highlighted in the re-submitted files.

Here are our responses/ corrections for your comments:

  1. Comment 1: Line 125-126 These numbers are few so how can the authors explain it? Did the authors ask them for how long they are addicted to it
  2. Response 1: Yes, one of the questions in the questionnaire addressed the duration of tobacco/nicotine product use (Table 2). One of the inclusion criteria required that the product be used for more than 2 years to exclude respondents who had only recently started using the product. A detailed explanation regarding the presence of Prevotella intermedia and Porphyromonas gingivalis in saliva will be provided in the discussion section. Please refer to the track changes in the resubmitted document.

  1. Comment 2: Line 152 what was the meaning of positive presence
  2. Response 2: Fusobacterium periodonticum was detected in all groups, with higher prevalence among cigarette smokers, nicotine pouch users, and electronic cigarette users. Since Fusobacterium periodonticum is commonly found in the oral microbiota, it may also be present in the saliva of control group respondents, as observed in this study (n=3). However, our results indicate that it is predominantly found in nicotine/tobacco users, suggesting a potential association with periodontal diseases.

  1. Comment 3: Comments for table 2 The authors can add alcohol intake and also biochemical test is needed to correlate the study, more data was needed.
  2. Response 3: Regular (daily) alcohol intake was an exclusion criteria for respondents to continue in the study, as it was considered that alcoholism constitutes a condition. Thank you for your comment; this factor will be added separately to the inclusion criteria for better clarity. Please check the track changes in the resubmitted document in the Materials and Methods section.

7.     Comment 4: First paragraph of discussion “It would be nice if the authors could discuss with numbers and the presence of harmful microorganisms”

8.     Response 4: Your comment has been taken into account. Please check the track changes in the resubmitted document.

9.     Comment 5: Line 184-187 Explain with another published article, try to cite more and comprehensively discuss.

10.  Response 5: Your comment has been taken into account. Please check the track changes in the resubmitted document.

11.  Comment 6: Line 207-208 what would be the possible reason for this

12.  Response 6: Your comment has been taken into account. Please check the track changes in the Discussion section in resubmitted document.
